# The *Caenorhabditis elegans* microtubule minus-end binding homolog PTRN-1 stabilizes synapses and neurites

Jana Dorfman Marcette*, Jessica Jie Chen, Michael L Nonet*

Department of Anatomy and Neurobiology, Washington University School of Medicine, St. Louis, United States

**Abstract** Microtubule dynamics facilitate neurite growth and establish morphology, but the role of minus-end binding proteins in these processes is largely unexplored. CAMSAP homologs associate with microtubule minus-ends, and are important for the stability of epithelial cell adhesions. In this study, we report morphological defects in neurons and neuromuscular defects in mutants of the *C. elegans* CAMSAP, *ptrn-1*. Mechanosensory neurons initially extend wild-type neurites, and subsequently remodel by overextending neurites and retracting synaptic branches and presynaptic varicosities. This neuronal remodeling can be activated by mutations known to alter microtubules, and depends on a functioning DLK-1 MAP kinase pathway. We found that PTRN-1 localizes to both neurites and synapses, and our results suggest that alterations of microtubule structures caused by loss of PTRN-1 function activates a remodeling program leading to changes in neurite morphology. We propose a model whereby minus-end microtubule stabilization mediated by a functional PTRN-1 is necessary for morphological maintenance of neurons.

## Introduction

In animals, stable neuronal morphology is necessary to maintain synaptic architecture. However, morphological alterations are also critical for plastic changes in synaptic strength, pruning of synaptic connections, and occasional responses to injury or other cues. Morphological reorganization of neurons is known to occur as a compensatory mechanism in cases of sight or hearing deprivation (*Karlen et al., 2006*; *Barone et al., 2013*), and medium spiny neurons are reported to experience reversible morphology changes in response to mild stress (*Bessa et al., 2013*). Neurons in *D. melanogaster* and *C. elegans*, as well as many mammalian peripheral neurons respond to injuries by remodeling their morphology including the growth of neurites, elimination of existing synapses, and establishment of new synapses (*Hilliard, 2009*).

The study of neuronal remodeling after injury in model organisms has elucidated cellular pathways responsible for morphological changes, however, little is known about the molecular cues that initiate these cascades. The conserved mitogen-activated protein (MAP) kinase kinase kinase DLK-1 functions to transmit regenerative signals in *C. elegans*, and is a component of regenerative signaling in organisms with more complex nervous systems (*Hammarlund et al., 2009*; *Xiong et al., 2010*; *Shin et al., 2012*). While there is a wealth of information about conserved down-stream signals affected by DLK-1 activity, very little information exits about how DLK-1 itself is activated (*Tedeschi and Bradke, 2013*). Recently, calcium spikes were shown to activate *C. elegans* DLK-1, but fly and mammalian homologs lack this calcium-binding domain (*Yan and Jin, 2012*). It is known that the conserved E3 ubiquitin ligase RPM-1 (PHR, PAM, HIGHWIRE) keeps levels of DLK-1 low (*Nakata et al., 2005*; *Xiong et al., 2010*). Axonal injury is reported to cause microtubule disassembly and disorganization of higher-order microtubule structures (*Erturk et al., 2007*). Microtubule depolymerization was recently found to activate DLK-1 signaling. However, this study did not investigate in what context (development,

*For correspondence:
marcette@pcg2.wustl.edu (JDM); nonetm@pcg2.wustl.edu (MLN)

**Competing interests:** The authors declare that no competing interests exist.

**Reviewing editor**: Oliver Hobert, Columbia University, United States

**eLife digest** Microtubules are tiny tubular structures made from many copies of proteins called tubulins. Microtubules have a number of important roles inside cells: they are part of the cytoskeleton that provides structural support for the cell; they help to pull chromosomes apart during cell division; and they guide the trafficking of proteins and molecules around inside the cell. Most microtubules are relatively unstable, undergoing continuous dis-assembly and re-assembly in response to the needs of the cell. The microtubules in the branches of nerve cells are an exception, remaining relatively stable over time. Now Marcette et al. and, independently, Richardson et al., have shown that a protein called PTRN-1 has an important role in stabilizing the microtubules in the nerve cells of nematode worms.

Marcette et al. became interested in the PTRN-1 protein after conducting a screen of randomly mutated worms to look for those with abnormally developed nerve cells. Although worms with a mutation in the gene encoding the PTRN-1 protein could form nerve cells that looked normal during early development, the pattern of branches on the nerve cells went awry later on. Moreover, the mutant worms lost the swellings that are normally found at the junctions between nerve cells, and they also moved their bodies in an odd way.

Engineering the mutant worms to produce the PTRN-1 protein in their nerve cells, but nowhere else, restored normal movement, and experiments with fluorescently tagged PTRN-1 proteins revealed that they are found on the microtubules within the nerve cells. Marcette et al. suggest that the microtubules become less stable when this protein is not present, and that this switches on a repair mechanism that remodels the nerve cells, providing that the active form of a protein called DLK-1 is present. Other mutations that reduce the stability of microtubules also triggered the same remodeling process. Future work will be necessary to uncover exactly what triggers the remodeling process, and to identify the other proteins that are involved in repairing the nerve cells.

regeneration, or degeneration) this microtubule-based mechanism might be functioning (*Bounoutas et al., 2011*).

Neurons contain both very stable and dynamic microtubule structures. These populations are maintained by the activities of cytoskeletal-associated proteins and by post-translational modifications (*Conde and Caceres, 2009*). Proteins known to be involved in stabilizing and organizing microtubule structures include molecular motors, and cross-linkers, as well as microtubule nucleating and severing proteins, and proteins involved in regulating plus-end dynamics (reviewed in *Subramanian and Kapoor, 2012*). Post-translational modifications to microtubules include acetylation, tyrosination, glutamylation, glycylation, and phosphorylation. These modifications affect the recruitment of microtubule-associated proteins and denote subcellular populations of microtubule structures such as the tyrosinated microtubules of dynamic growth cones or the acetylated microtubules of stable neurites (*Baas and Black, 1990*; *Brown et al., 1993*; *Janke and Bulinski, 2011*; *Garnham and Roll-Mecak, 2012*). In *C. elegans*, microtubule acetylation mutants display short microtubules, alterations in microtubule protofilament number, and association of filaments into higher-order structures (*Cueva et al., 2012*; *Topalidou et al., 2012*). Furthermore, microtubule de-acetylation was recently shown to be an early signal for the development of growth-cones in neuronal remodeling associated with regeneration (*Cho and Cavalli, 2012*). It is currently unclear how destabilizing modifications or depolymerization of microtubule structures could activate neuronal remodeling. Depolymerization events that occur in dynamic microtubule populations clearly do not cause neurite remodeling in healthy neurons, so it is not likely that depolymerization itself or free microtubule monomers are the remodeling signal.

One possibility is that the normally stable minus-ends of microtubules initiate remodeling signals. In this study, we report that genetic lesions in *ptrn-1*, the *C. elegans* homolog of the microtubule minus-end binding calmodulin-regulated spectrin-associated (CAMSAP) family cause neurite remodeling including overextension of neurites and retraction of collateral presynaptic varicosities and branches. These remodeling events require DLK-1, a known player in morphology changes associated with degenerative and regenerative signaling in neurons. This is the first report of a CAMSAP family member being involved in remodeling of neurite morphology. Members of the CAMSAP family are known to associate with microtubule minus ends and stabilize microtubule structures (*Goshima et al.,*

*2007*; *Meng et al., 2008*; *Baines et al., 2009*; *Goodwin and Vale, 2010*; *Tanaka et al., 2012*). CAMSAPs are found in mammalian neuronal tissue (*Baines et al., 2009*; *Guo et al., 2012*; *Yamamoto et al., 2009*), and genetic variants of human CAMSAP1L1 were recently shown to confer susceptibility to epilepsy (*Guo et al., 2012*). Thus understanding the neuronal consequences of disruptions to CAMSAP function may begin to inform our understanding of disease mechanisms.

We found that disruptions in *ptrn-1* cause alterations to neuronal morphology, and these changes occur via a DLK-1 neuronal remodeling program. We were able to mimic and enhance *ptrn-1* loss-of-function phenotypes with genetic and pharmacological alterations of the microtubule cytoskeleton. Furthermore, mutation of *dlk-1* suppressed a microtubule acetylation mutant supporting the idea that altering microtubule structures is an upstream signal for DLK-1 activation. PTRN-1 is present in neurites and at sites of synaptic connectivity, suggesting local microtubule minus-end regulation may affect these mechanisms. In addition to identifying a novel function for CAMSAPs in the maintenance of neuron morphology, our results suggest that alterations to microtubule structures caused by loss of CAMSAP function activates a neuronal remodeling program. We propose a model whereby local minus-end microtubule stabilization mediated by CAMSAPs is necessary for the stabilization of mature neurite morphology.

## Results

### Identification of the *C. elegans* CAMSAP family homolog *ptrn-1* in a screen for altered neuronal morphology

We use *C. elegans* touch receptor neurons (TRNs) to study establishment and maintenance of neuronal morphology. The laterally positioned TRN neurons each extend a neurite in which touch receptors are distributed at regular intervals (*Zhang et al., 2004*). Together PLM and ALM lateral neurites cover the length of the animal, but they do not overlap (*Figure 1A*). ALM and PLM chemical synapses are on collateral branches that terminate in presynaptic varicosities. Each PLM mechanosensory neuron forms a single presynaptic varicosity in the ventral nerve cord just posterior to the vulva (*Figure 1A*). Thus, the lateral TRNs have a highly stereotyped morphology that allows visualization of individual neurons in living animals.

We screened 1134 haploid genomes looking for maintenance defects in the presynaptic varicosities of PLM mechanosensory neurons. Three candidates from this screen remain uncharacterized, the fourth, *js1286*, is the focus of this work. We identified *js1286* as a nonsense mutation in the *C. elegans* CAMSAP, *ptrn-1*, that caused a range of partially penetrant phenotypes in PLM and ALM neurons (*Figure 2A,B*, *Figure 1—figure supplement 1*). Defects seen in PLM neurons included loss of collateral branches and presynaptic varicosities, as well as overextended neurites that exited the lateral cord and formed ectopic presynaptic varicosities in the ventral nerve cord anterior to the vulva (*Figure 1B,C*). We observed the same phenotypes in an independently generated deletion allele, *ptrn-1 (tm5597)* (*Figure 2A, 1C*, *Figure 1—figure supplement 1*), indicating the observed defects in *ptrn-1* are representative of the *ptrn-1* null phenotype. We also observed morphological defects in motor neurons and command interneurons, and behavioral defects in locomotion indicating that PTRN-1 functions broadly in the nervous system (*Figure 1D,E*, *Figure 1—figure supplement 2*).

### PTRN-1 functions in neurons

Changes in neuronal morphology could be caused by cell autonomous alterations or by changes in surrounding tissues. We noticed that *ptrn-1* animals have uncoordinated locomotion and body positioning, a characteristic shared by mutants with defects in either neurons or muscles. Prior work has focused mostly on the role of CAMSAPs in epithelial tissue; however by expressing a *ptrn-1* transcriptional reporter construct, we found that *ptrn-1* is broadly expressed in the nervous system (*Figure 3*). Driving *ptrn-1:mcherry* under the pan-neuronal (*rab-3*) promoter, but not the pan-muscle (*myo-3*) promoter rescued PLM morphology defects indicating that PTRN-1 functions in neurons to regulate PLM morphology (*Figure 1C*).

The *ptrn-1* gene represents the only *C. elegans* ortholog of the CAMSAP family. The domain structure for CAMSAPs consists of an N-terminal calponin-homology domain, a C-terminal microtubule-binding domain and an intervening region of predicted coiled-coil domains (*Figure 2B*). CAMSAPs regulate microtubule minus-end depolymerization, and are important for the stability of epithelial cell adhesions (*Goodwin and Vale, 2010*; *Meng et al., 2008*). CAMSAPs localize to microtubule minus ends,

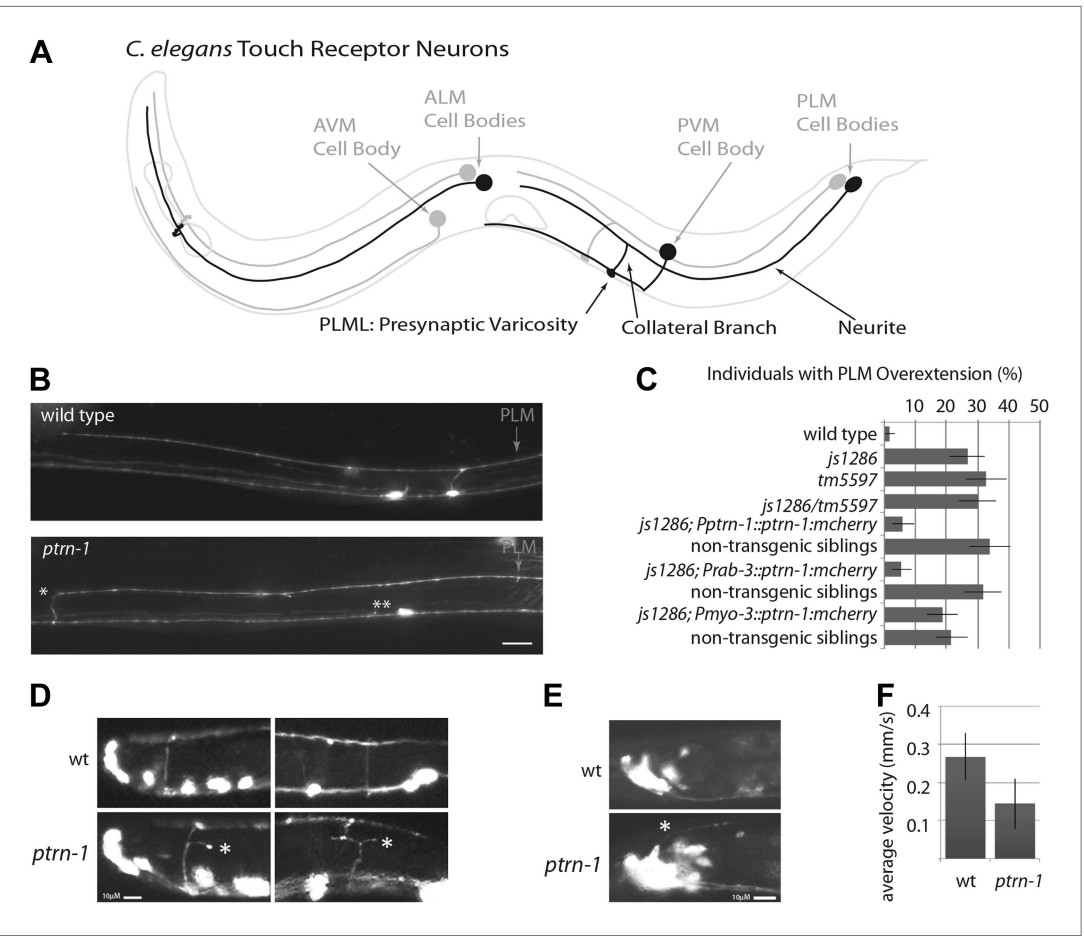

**Figure 1**. *ptrn-1* mutant neurite overextension and collateral branch retraction phenotypes. (**A**) Schematic diagram of an L4 *C. elegans* hermaphrodite showing the morphology of the touch receptor neurons. Neurons on the left side are in black, and those on the right side are in gray. (**B**) Morphology of mechanosensory neurons in wild-type and *ptrn-1* mutants, as seen by expression of mRFP under the control of the *mec-7* promoter. Defects in *ptrn-1* animals include PLM neurite overextension and ectopic formation of presynaptic varicosities (*), retraction of PLM collateral presynaptic varicosities and branches (**). Scale bar = 10 μm. Additional examples and variance in Ptrn-1 phenotypes are presented in *Figure 1—figure supplement 1*. (**C**) Quantification of the PLM neurite overextension phenotype at 22°C. Both *ptrn-1* alleles (*js1286* and *tm5597*) show similar defects, as do trans-heterozygotes. Also shown is the rescue of the PLM neurite overextension phenotype by expression of PTRN-1a-mCherry under a native promoter and in neurons under a *rab-3* promoter, but not under a *myo-3* promoter. Non-transgenic siblings were identified as individuals from mothers segregating the transgene, but that lacked detectable mCherry fluorescence. n = 50–60; error bars = SEP. (**D**) Aberrant formation and extension of neurites (*) was found in D-type motor neurons. Quantification is in *Figure 1—figure supplement 2*. (**E**) Aberrant formation and extension of neurites (*) was found in command interneurons. Quantification is in *Figure 1—figure supplement 2*. (**F**) Quantification of locomotory defects using L4 animals. Error bars represent standard deviation. N = 6.

The following figure supplements are available for figure 1:

**Figure supplement 1**. Variance in the *ptrn-1* mutant TRN phenotypes.

**Figure supplement 2**. Quantification of other morphology phenotypes in *ptrn-1* mutants.

and in puncta distributed along higher-order microtubule structures (*Baines et al., 2009*; *Goodwin and Vale, 2010*; *Meng et al., 2008*; *Tanaka et al., 2012*). Consistent with these findings, we see that both PTRN-1-cherry and GFP-PTRN-1 constructs exhibit a punctate distribution along neurites and at sites of high synaptic connectivity including the nerve-ring and ventral nerve cord (*Figure 4*). In neuronal cell bodies, PTRN-1-mcherry and GFP-PTRN-1 could be found in one or two puncta

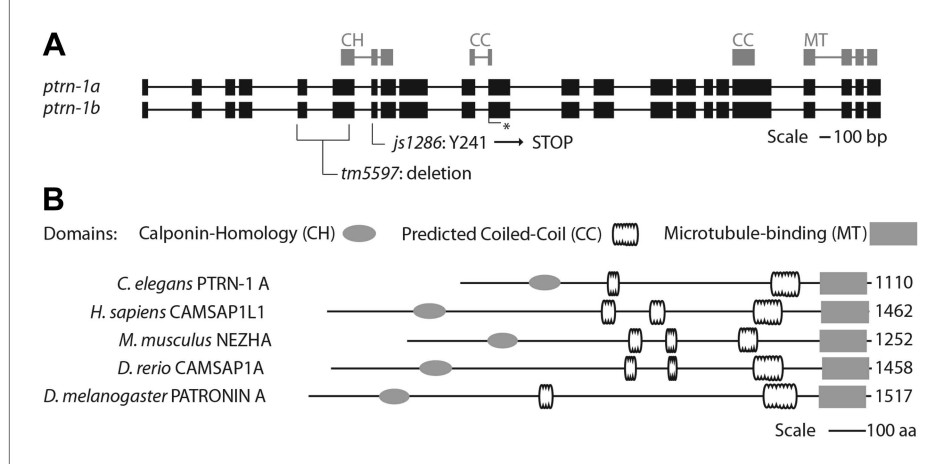

**Figure 2**. Gene and protein structure of *C. elegans ptrn-1*. (**A**) Exon (thick regions) and intron (thin regions) organization of the *ptrn-1* locus. The position of structural domains is demarcated above and the position of the genetic lesion for each of the *ptrn-1* alleles is shown below. *ptrn-1b* differs from *ptrn-1a* by the inclusion of 6 bp (2 amino acids) at the exon 10–11 boundary (*). Nucleotide sequences for *ptrn-1a* and *ptrn-1b* are listed in *Figure 2—figure supplements 1 and 2*, respectively. Omitted from the figure is a distinct *ptrn-1* isoform lacking the C-terminal half of the protein, which is not represented in RNA-seq data for which the sole support is a single compound EST yk1268c03. (**B**) The CAMSAP protein family consists of a calponin-homology domain followed by one or two 30–40 amino acid coiled-coil regions, a 70–85 amino acid coiled-coil region and a CKK microtubule-binding domain. CAMSAP sequences from *C. elegans* (this paper), *D. melanogaster* (AAO41362.1), *D. rerio* (NP_001159727.1), and *H. sapiens* (AAI25231.1) were used for homology analysis. Domains were annotated by running sequences through the NCBI Conserved Domain Database. Coiled-coil regions were defined as those predicted by both the Paircoil2 and COILS programs.

The following figure supplements are available for figure 2:

**Figure supplement 1**. Nucleotide sequence of *ptrn-1a*.

**Figure supplement 2**. Nucleotide sequence of *ptrn-1b*.

(*Figure 4A–C,E–G*), and colocalization of GFP-PTRN-1 with the mechanosensory-specific ß-tubulin MEC-7 was found (*Figure 4H–I*). These results are consistent with *C. elegans* PTRN-1 functioning in neurons.

## *ptrn-1* mutants fail to maintain wild-type neuronal morphology

We wanted to distinguish between defects that arise during development, and those that are a result of a failure to maintain or stabilize normally patterned neurons. Although ALM neurites extend during embryonic development, the development of PLM neurons occurs in several distinct steps some of which occur during the L1 larval stage (*Figure 5A*). After extending lateral neurites in the embryo, PLMs form collateral branches early in the L1 larval stage, and subsequently active zone synaptic machinery, vesicles, and mitochondria are recruited successively to nascent collateral presynaptic varicosities. The overall morphological pattern is maintained as the animal quadruples its body length from the L1 to L4 larval stage, with the only substantive change being a ten-fold increase in the size of presynaptic varicosity. In *ptrn-1* animals, we noticed that wild-type morphology is lost after the initial L1 patterning, with animals missing presynaptic varicosities and overextending neurites (*Figure 5B*). A small fraction of *ptrn-1* animals had overextended neurites and/or missing presynaptic varicosities at the L1 stage (*Figure 5B*); however from this data set, we are unable to determine if these animals originally formed morphologically normal neurites and experienced early retraction/overextension defects prior to the L1 time-point or if they represent a partially penetrant role for *ptrn-1* in initial L1 patterning. We also examined individual animals at both the L1 and L4 larval stages to confirm that aberrant morphology phenotypes do occur subsequent to L1 developmental patterning (*Figure 5C*). Ectopic presynaptic varicosities that formed at the terminus of overextended neurites in L1 larva were

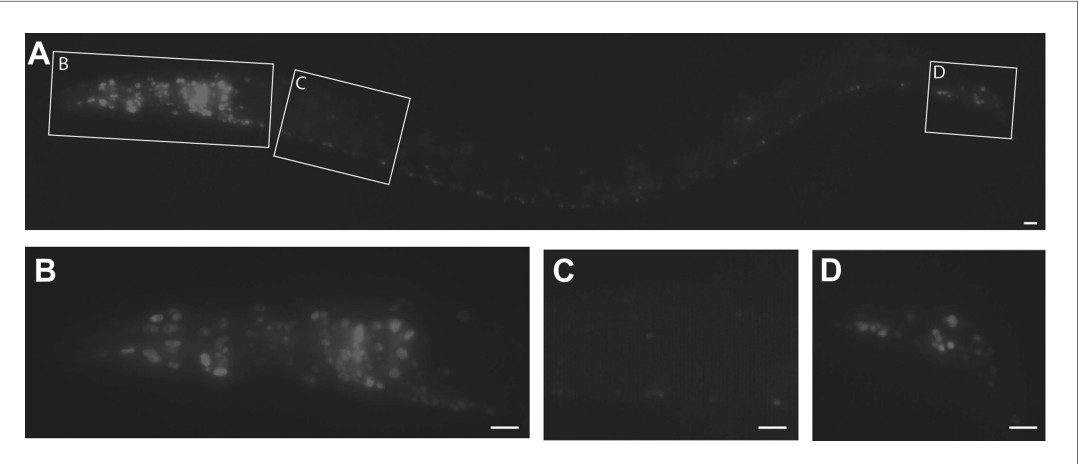

**Figure 3**. *C .elegans ptrn-1* is broadly expressed in neuronal tissue. (**A**) *ptrn-1* expression pattern from a transcriptional fusion between mCherry and a *ptrn-1* promoter segment sufficient to rescue Ptrn-1 TRN cellular phenotypes (see *Figure 1C* for rescue). 0.8 kb was chosen as the promoter fragment because of the presence of a 1.2 kb Mariner transposable element at this position 5′ of the *ptrn-1* ATG. Scale bars = 10 µm. (**B**–**D**) Views at higher magnification of the head neuronal ganglion (**B**), ventral nerve cord (**C**), and pre-anal and lumbar neuronal ganglia (**D**). mCherry is primarily nuclear localized.
The following figure supplements are available for figure 3:

**Figure supplement 1**. Close-up view of the *ptrn-1* expression pattern.

also subject to retracting; further supporting the idea that *ptrn-1* is required for maintenance, rather than initial establishment of neurite morphology.

The average size of presynaptic varicosities that persist in *ptrn-1* animals not significantly different from wt (10 ± 3 µM² for *ptrn-1* compared to 12 ± 3 µM² for wild-type N = 20; p=0.06). While presynaptic varicosity retraction is partially penetrant, once initiated, retraction goes to completion. Thus, maintenance of presynaptic varicosities seems to be subject to a binary switch, and this suggested to us that loss of *ptrn-1* might result in initiation or transmission of a retraction signal.

## PTRN-1 acts upstream of the DLK-1 neuronal remodeling pathway

The Ptrn-1 phenotypes of neurite overextension and the absence of collateral presynaptic varicosities are reminiscent of defects in *C. elegans* caused by loss of function mutations in the ubiquitin ligase *rpm-1*, a known regulator of neurite remodeling (*Schaefer et al., 2000*; *Zhen et al., 2000*). Loss of *rpm-1* results in activation of the DLK-1 MAP kinase pathway and *rpm-1* phenotypes are DLK-1 dependent (*Nakata et al., 2005*). Because of the phenotypic similarities between *rpm-1* and *ptrn-1* mutants, we wondered if *dlk-1* is required for the remodeling events in *ptrn-1* mutants. We found that both *rpm-1* and *ptrn-1* L4 animals have similar PLM morphological defects, including neurite overextensions that target the ventral nerve cord, and the absence of collateral presynaptic varicosities (*Figure 6A,B*). By contrast, *dlk-1* loss-of-function mutants exhibit grossly normal PLM morphology. Furthermore, like *rpm-1*, we found that *ptrn-1* presynaptic varicosity retraction and overextension defects are suppressed by loss of *dlk-1*, as well as in a triple-mutant background (*Figure 6A,B*). However, we noticed that in double and triple mutants usually one or both of the varicosities were smaller and elongated (*Figure 6B*), indicating additive roles for *dlk-1*, *ptrn-1* and *rpm-1* in shaping the synapse. Our data show that *ptrn-1* functions upstream of *dlk-1*, and indicates that loss of PTRN-1 activates a *C. elegans* neuronal remodeling program.

## CAMSAPs likely function via microtubule regulation In neurons

Remodeling pathways may be activated through a microtubule-based mechanism in *ptrn-1* mutants. Recently, microtubule depolymerization was reported to activate DLK-1, and ß-tubulin mutants have a reduced capacity to regenerate despite the fact that neurite extension appears wild type in these animals (*Bounoutas et al., 2011*; *Kirszenblat et al., 2013*). PTRN-1 homologs have been shown to

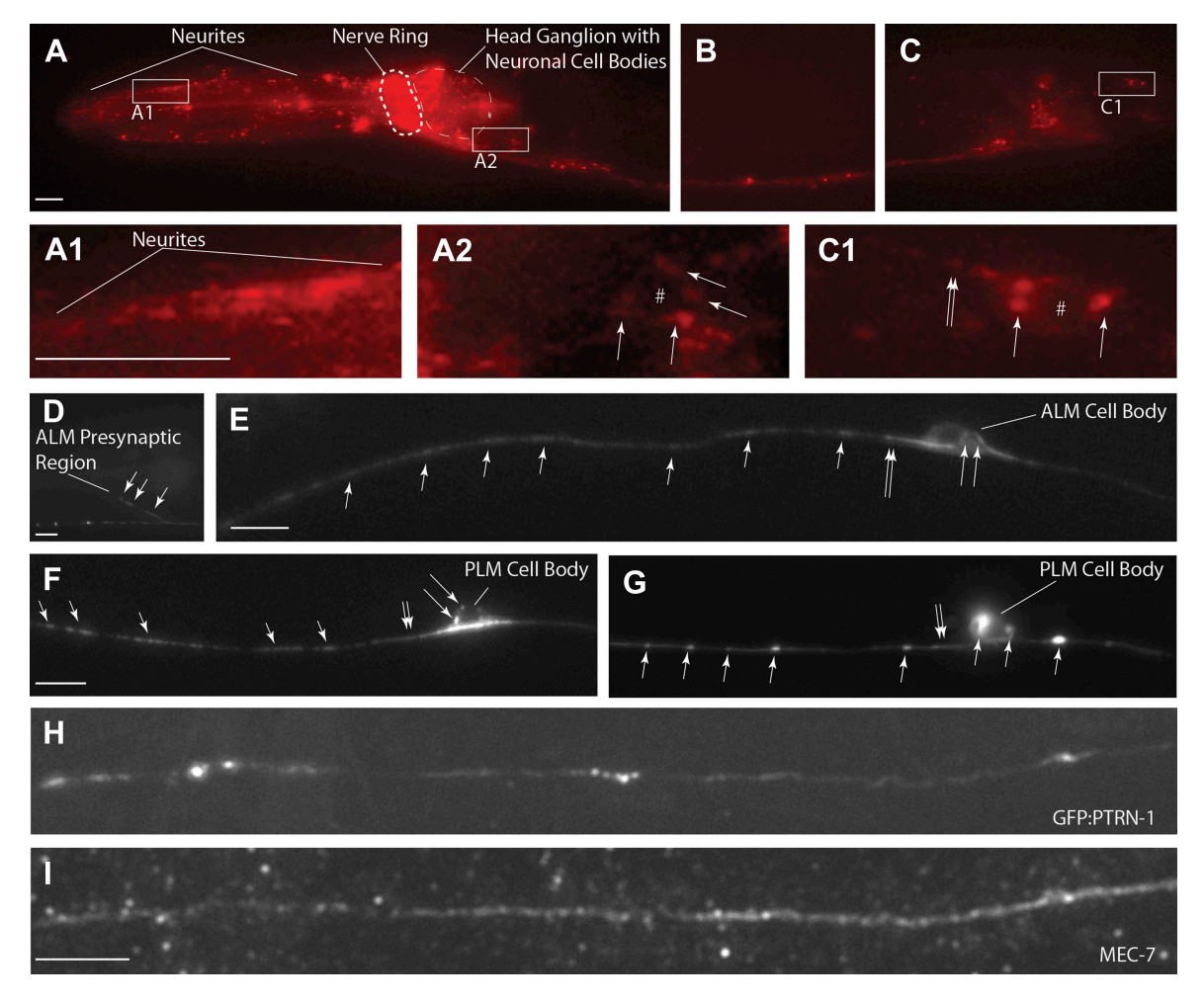

**Figure 4**. PTRN-1-localization pattern. Expression of PTRN-1-mcherry under a *ptrn-1* 0.8 kb promoter segment sufficient to rescue Ptrn-1 TRN cellular phenotypes (**A–C**) (see *Figure 1C* for rescue data). Enlarged views of neurites (**A1**), a head ganglion cell body (**A2**), and a tail neuron with isolated morphology (**C1**) are shown. Puncta (arrows), nuclear exclusion (*), and neurite (double arrow) are noted. Single-cell resolution showing a N-terminal GFP-PTRN-1 fusion protein under the control of the TRN *mec-7* promoter in ALM and PLM neurons (**D–G**). Puncta (arrows) and neurites (double arrows) are noted. Scale bars = 10 µm. Whole-mount immunofluorescent staining with anti-GFP (**H**) and anti-MEC-7 (**I**) antibodies. Co-localization of GFP-PTRN-1 with MEC-7 in mechanosensory neurons is shown.

associate with microtubule minus ends and to regulate microtubule dynamics in ex vivo and in vitro model systems (*Baines et al., 2009*; *Goodwin and Vale, 2010*; *Goshima et al., 2007*; *Meng et al., 2008*; *Tanaka et al., 2012*). It is therefore possible that loss of *ptrn-1* alters microtubule structures, initiating a DLK-1-based remodeling program. Because investigating neuronal microtubule dynamics in *C. elegans* neurons in vivo is technically difficult, we explored a link with microtubule structures by looking for Ptrn-1 phenotypes using known genetic and pharmacological manipulations that perturb the microtubule cytoskeleton.

We monitored PLM neurite overextension after exposing worms to the microtubule-poisoning drug colchicine. Colchicine is known to destabilize microtubules (*Gigant et al., 2009*), and while colchicine treatment did not cause regenerative phenotypes in wild-type worms, it caused these phenotypes to be more penetrant in *ptrn-1* mutants (*Figure 7A*). Because the effects of colchicine treatment were additive in combination with *ptrn-1* mutants, it is possible that loss of PTRN-1 affects microtubule structures, similar to CAMSAPS in other systems. However, we cannot rule out the possibility that although *ptrn-1* mutants are more susceptible to sub-threshold amounts of colchicine-induced perturbation, the effect is independent of direct *ptrn-1* alterations in microtubule structures.

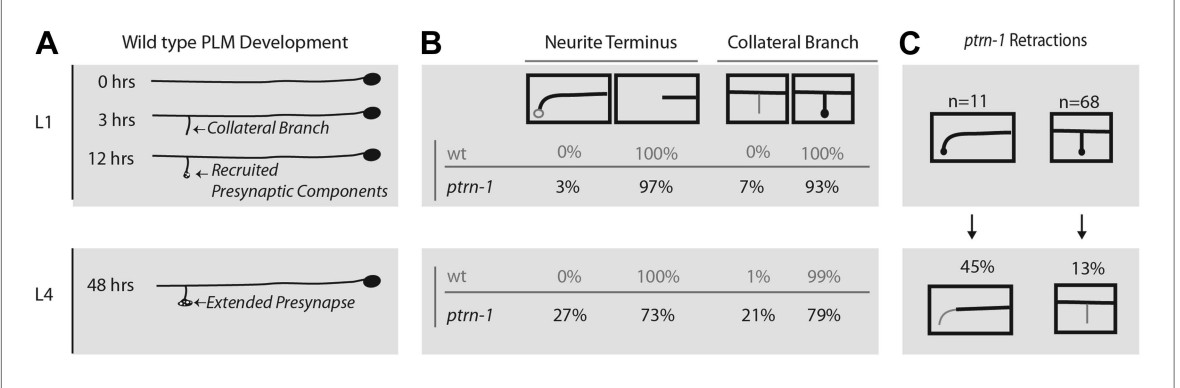

**Figure 5**. *ptrn-1* mutants fail to stabilize mature PLM morphology and synaptic contacts. (**A**) Schematic diagram illustrating wild-type PLM neuron developmental changes during the L1 larva from 0 to 12 hr after hatching, as well as the structure of PLM neurons in L4 larva—note that the figure is not drawn to scale. PLM neurons maintain their relative position within the animal from L1 to L4, but L4 are approximately four times longer than L1 animals. (**B**) Changes observed in PLM neuronal morphology in wild-type and *ptrn-1* (*js1286*) mutants from a synchronized population of animals analyzed at L1 (10 hr) and L4 (36 hr) stages. Animals were grown at 22°C. PLM neurons show both overextension and ventral targeting of the anterior neurite in *ptrn-1* mutants, but rarely in wild type. The ventral targeting posterior neurites display variable sized varicosities that are labeled with presynaptic components. *ptrn-1* animals also often exhibit loss of collateral branch and associated presynaptic structures. N = 90–98 neurites. (**C**) Retraction of collateral branch synaptic varicosities and overextended ectopic ventral neurite targeted varicosities in animals imaged as both L1 and L4 larva. N = 11 and 68 neurites. (**B** and **C**) Black lines and circles represent neurites and presynaptic varicosities respectively. Gray lines and open circles represent variability in *ptrn-1* animals.

We also found that colchicine treatment suppressed the formation of ALM posterior neurites (data not shown), as has been reported when these structures have occurred in other mutants (*Kirszenblat et al., 2013*; *Topalidou et al., 2012*; *van Zundert et al., 2002*). While abnormal lateral extension of ALM and PLM neurites occurs in a variety of mutants (*Kirszenblat et al., 2013*; *Mohamed et al., 2012*; *Topalidou et al., 2012*; *Tulgren et al., 2011*), the Ptrn-1 PLM phenotype, where overextended neurites migrate to the ventral nerve cord just anterior to the vulva has only been found in mutants that undergo neurite remodeling. Microtubule acetylation mutants alter microtubules and are reported to

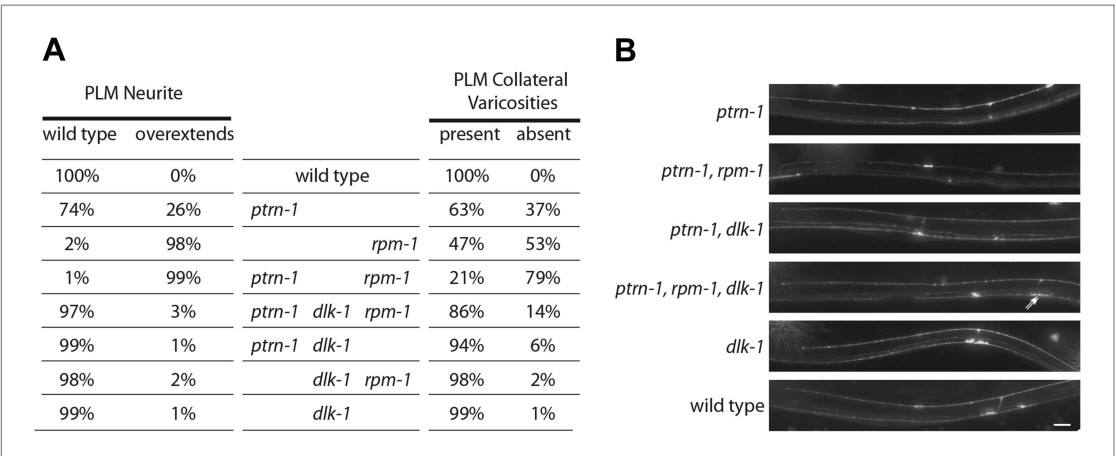

**Figure 6**. *dlk-1* MAP kinase mutants suppress Ptrn-1 overextension phenotypes. (**A**) Single, double, and triple mutant combinations of *ptrn-1*(*tm5597*), *dlk-1*(*km12*) and *rpm-1*(*ok364*) were grown at 22°C, scored as L4 larvae for PLM neurite morphology defects using *jsIs973*(P*mec-7*:mRFP), and for the presence of collateral branch presynaptic varicosities using *jsIs821*(P*mec-7*::GFP-RAB-3). Since PVM forms small GFP puncta in the ventral nerve cord (see *Figure 1A,B*), PLM collateral varicosities which retained a small GFP-RAB-3 puncta in the ventral nerve cord cannot be unambiguously distinguished from PLM collaterals completely lacking presynaptic components. n = 200 PLM cells from 100 animals. (**B**) Midbody of L4 animals expressing mRFP in TRNs under P*mec-7* (*jsIs973*) showing the wild-type and mutant PLM morphology associated with *ptrn-1* and *dlk-1*. Elongated varicosity in *ptrn-1*, *rpm-1*, *dlk-1* triple mutant identified with double arrows. Scale bar = 10 μm.

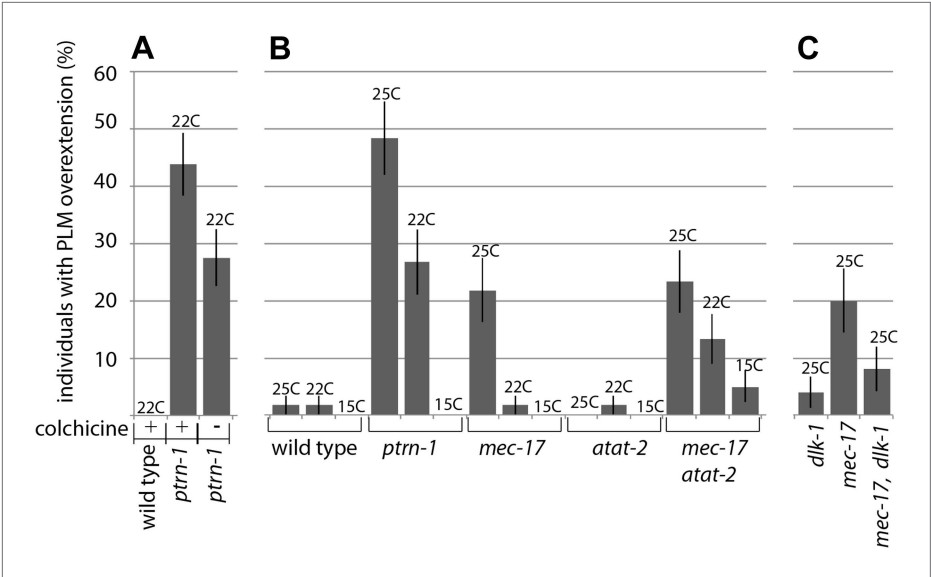

**Figure 7**. Manipulating the microtubule cytoskeleton leads to terminal neurite extension and ventral nerve cord targeting. (**A**) Wild-type and *ptrn-1*(*js1286*) mutants were grown for two generations on plates containing 1 mM colchicine and scored for PLM neurite overextension (n = 80). Error bars represent SEP. (**B**) Wild-type, *ptrn-1* mutants and mutants with lesions in microtubule acetylases (paralogs *mec-17* and *atat-2*) were scored for PLM neurite overgrowth defects at various temperatures (n = 55-60). Error bars represent SEP. (**C**) *dlk-1* suppresses the PLM neurite overextension phenotypes of *mec-17* (n = 50). Error bars represent SEP.

have these ALM secondary posterior neurites (*Cueva et al., 2012*; *Topalidou et al., 2012*). We asked if disruptions caused by altered microtubule acetylation would be sufficient to cause remodeling phenotypes. We found that the microtubule acetylation mutant *mec-17* caused PLM neurite overextension; furthermore, like *ptrn-1* mutants, this phenotype was suppressed by growing the worms in cooler temperatures, and in a *mec-17*, *dlk-1* double mutant (*Figure 7B,C*). These results indicate that altering the microtubule cytoskeleton can activate a DLK-1 remodeling program. We also saw absence of collateral presynaptic varicosities in *mec-17* mutant animals, however this phenotype was additive in *mec-17*, *dlk-1* double mutants indicating that *mec-17* and *dlk-1* have parallel presynaptic roles (data not shown).

Our work using genetic and pharmacological manipulations of the cytoskeleton supports the idea that altering microtubules activates neuronal remodeling programs, and suggests that the CAMSAP PTRN-1 acts in this pathway. Similar to the function of CAMSAPs in other systems, the neuronal function of CAMSAP proteins may involve the regulation of microtubules or microtubule structures.

## Discussion

Alterations in neuronal morphology are associated with human developmental and neurodegenerative diseases (*Newey et al., 2005*; *Lin et al., 2009*; *Liu, 2011*; *Goellner and Aberle, 2012*; *Yu and Lu, 2012*) and it is therefore important to understand the cellular mechanisms that underlie these changes. The conserved DLK-1 kinase cascade promotes changes in morphology associated with neurite remodeling (*Hammarlund et al., 2009*), but how this cascade is activated has remained elusive. Our data show that loss of the conserved CAMSAP PTRN-1 induces DLK-mediated remodeling phenotypes. Because CAMSAP family members are known to associate with microtubule minus ends and regulate microtubule structures (*Goodwin and Vale, 2010*; *Meng et al., 2008*; *Tanaka et al., 2012*), we propose a model where microtubule structures act as upstream sensors to neuronal programs that lead to synapse elimination and growth of neurites.

How does loss of PTRN-1 initiate neuronal remodeling? PTRN-1 could itself be a part of the machinery that represses the DLK-1 pathway, or it could be acting indirectly through the sequestration of a remodeling sensor. Because we can enhance defects in *ptrn-1* mutants with the microtubule-binding drug colchicine, and mimic Ptrn-1 phenotypes with microtubule acetylation mutants, PTRN-1 is not likely to be the signal itself. We favor a model where perturbations of the microtubule cytoskeleton

are responsible for activation of remodeling in *ptrn-1* mutants. Interestingly, recent findings indicate that at cooler temperatures, neuronal microtubules incorporate microtubule monomers that have been stabilized through post-translational modification (*Song et al., 2013*). It is possible that the rescue we see of Ptrn-1 defects by growing animals in cooler temperatures is due to this mechanism; by incorporating cold-stable monomers, microtubule structures may be less sensitive to cytoskeletal alterations caused by *ptrn-1* or acetylation mutants.

Altered association of microtubule filaments into networks may initiate neurite remodeling rather than disruption of individual microtubules. Separation of microtubule filaments from organized cytoskeletal networks has been reported in experiments that decrease expression of PTRN-1 homologs and in microtubule acetylation mutants (*Goodwin and Vale, 2010*; *Topalidou et al., 2012*). Such alterations would cause the release of factors associated with tethering connections between microtubule filaments, as well as the appearance of free minus ends, either of which could be a signal that a remodeling response is needed.

In *ptrn-1* mutants, regenerative changes in neuron morphology occurred after developmental patterning of the nervous system. Potential explanations for this observation include that regenerative DLK-1 signaling pathways only become competent to respond to activating cues after development is complete or that the regenerative cue appears later, perhaps because CAMSAPs stabilize microtubule structures present only in mature neurons. One possibility is that CAMSAPs are involved in microtubule-based sequestration of a RPM-1 inhibitor. RPM-1 is known to keep levels of DLK-1 low by targeting it for degradation, and this mechanism must be surmounted in *ptrn-1* mutants that have a functional *rpm-1* gene. Alternatively, if DLK-1 or a DLK-1 activator is being sequestered, then perhaps a build-up of these stores over time overwhelms the ability of RPM-1 to keep DLK-1 levels low.

Neurites contain overlapping microtubules that must be interconnected to fulfill their known functions in trafficking and morphological integrity. These higher-order cytoskeletal networks must be organized in a way that stabilizes microtubule minus ends, in addition to linking individual microtubules together. The minus-end of microtubules are both known for durability and are distributed at regular intervals throughout neurites (*Baas and Black, 1990*; *Kurachi et al., 1999*; *Cueva et al., 2007*; *Bellanger et al., 2012*). We found PTRN-1 to be distributed in puncta along neurites, and these structures thus represent a potential reservoir for sensors that promote remodeling. Although these results indicate that neuronal CAMSAPs function similarly to epithelial CAMSAPs by stabilizing microtubule networks, additional work will be required to explore the link between *ptrn-1* disruption and specific changes to microtubule-based structures.

We also found evidence for PTRN-1 function outside of neuronal remodeling pathways, including roles in neurite polarity and in sculpting the shape of synapses. The appearance of ALM posterior neurites, seen in *ptrn-1* mutants, is thought to be a neuron polarity defect (*Hilliard and Bargmann, 2006*; *Kirszenblat et al., 2013*; *Prasad and Clark, 2006*), and the cell body puncta seen in animals expressing PTRN-1:mcherry may be microtubule organizing centers necessary for polarized neurite outgrowth. However, the composition of ectopic ALM posterior neurites is incompletely understood. These extensions are reported to occur due to hyper-stabilization of microtubules (*Kirszenblat et al., 2013*), but also appear in mutants where microtubules and cytoskeletal networks are disrupted (*Topalidou et al., 2012*). Mature neurons are known to have stable microtubule structures due to post-translational modifications and interactions with cytoskeletal associated proteins (*Conde and Caceres, 2009*; *Kawataki et al., 2008*; *Song et al., 2013*). The differential effects of colchicine that we see on ALM and PLM phenotypes in *ptrn-1* mutants may occur because of the differences in microtubule-based structures present in ectopic (ALM) vs native (PLM) neurites. Future work will be necessary to probe the role of CAMSAPs in neuron polarity.

In *ptrn-1*, *dlk-1* double mutants, we observed small and sometimes elongated presynaptic varicosities. It was recently reported that the *D. melanogaster* homolog of DLK-1 has developmental roles in synaptic structure that are independent from roles in neuronal remodeling (*Klinedinst et al., 2013*), and our data support similar conclusions for presynaptic varicosities in *C. elegans*. Synapses are specialized cell adhesion sites, and proteins involved in cellular adhesions have been found to be involved in maintenance of synaptic structures (*Chang and Balice-Gordon, 2000*; *Pielage et al., 2008*; *Geissler et al., 2013*; *Pielarski et al., 2013*). CAMSAPs mediate connections between microtubule minus ends and adhesion proteins at epithelial adherens junctions (*Meng et al., 2008*), and the synaptic-localized CAMSAPs we observed may play a similar bridging role at synaptic adhesion sites.

Our data represent the first characterization of CAMSAP function in an intact animal. In *C. elegans*, we detected CAMSAP expression primarily in neurons. Mammalian CAMSAPs have also been found

in neuronal tissue, and CAMSAP1L1 is a genetic trait locus for epilepsy (*Guo et al., 2012*). Abnormal neuronal patterning and seizure-induced morphological remodeling are both thought to contribute to continuing seizures and the development of epilepsies (*Houser et al., 2012*; *O'Dell et al., 2012*; *Zhao and Overstreet-Wadiche, 2008*). We found that mutations in the *C. elegans* CAMSAP homolog, *ptrn-1* initiate remodeling programs that cause aberrant extension and morphological re-patterning of neurites. Our data indicate that cytoskeletal-based activation of regenerative programs may exist in the absence of external injury signals. We speculate that without the stabilizing function of CAMSAP proteins, neuronal remodeling may be more likely to occur, a phenomenon we observed in *C. elegans ptrn-1* mutants. We further speculate that the function of CAMSAPs in maintenance of neuronal morphology may be responsible for the association of CAMSAP1L1 with epilepsy.

This work represents the first evidence of a role for microtubule minus-ends in neuron remodeling pathways. It identifies CAMSAPs as regulators of neuronal morphology, positions both CAMSAPs and microtubule structures as early regulators of neuronal-remodeling programs, and suggests that CAMSAPS regulate microtubule organization in neurons. Additionally, we found evidence for CAMSAP function in establishing neuron polarity and stabilizing the morphology of the synapse. Future work will be necessary to uncover the specific nature of the regenerative signal activating DLK-1 pathways, and to identify the CAMSAP-interacting proteins that function in neurons.

## Materials and Methods

### Strains and genetics

Strains were maintained at 22°C, unless otherwise specified, on Nematode Growth Medium (NGM) agar plates spotted with OP50 *E. coli* (*Brenner, 1974*). Some strains were provided by the Caenorhabditis Genetics Center, which is funded by NIH Office of Research Infrastructure Programs (P40 OD010440). The *tm5597* deletion mutation was provided by the MITANI Lab through the National Bio-Resource Project of the MEXT, Japan.

### Mutagenesis and screening

The *ptrn-1*(*js1286*) allele was isolated from an ENU-mutagenesis screen of animals carrying the *jsIs973* and *jsIs1077* transgenes. *js1286* was determined to be a recessive lesion mapping to the X chromosome during outcrossing and positioned to the right of the single nucleotide polymorphism (SNP) Ce6-1456 (X:+17) using standard SNP mapping using the HA-8 wild Hawaiian *C. elegans* strain (*Davis et al., 2005*). Whole genome sequencing by Robert Barstead was performed at Oklahoma Medical Research Foundation and the data set was analyzed using Whole Genomes (http://seqreport.omrf.org/genome/), a web-based alignment and analysis program. Approximately 30-fold coverage using 100 bp paired end reads revealed only 1 homozygous missense and 1 homozygous non-sense lesion in coding regions right of uCE6-1456. Fosmid rescue with a 38-kb genomic region (WRM0636bB07) containing F35B3.5a (*ptrn-1*; a.k.a. *cspr-1* and *pqn-34*) suggested the lesion in this gene was the causative lesion resulting in the axon overgrowth phenotype.

### Plasmid construction

Oligonucleotides used in plasmid construction are listed in *Supplementary file 1*.

NM2498 pCFJ151 Prab-3. The *rab-3* promoter was amplified using oligonucleotides 4216 and 4217, digested with AflII and SbfI and inserted into pCFJ151 (*Frokjaer-Jensen et al., 2008*).

NM2703 pCFJ151 Prab-3::mCherry::MCS. mcherry was amplified from pCFJ104 (*Frokjaer-Jensen et al., 2008*) using 4218 and 4533, digested with XhoI and AvrII and inserted into similarly digested NM2498.

NM2704 pCFJ151 Prab-3::mCherry::MCS::unc-10 3'UTR. The 3' UTR region of *unc-10* was amplified using oligonucleotides 4651 and 4652, digested with BsiWI and SgrAI and inserted into similarly digested NM2703.

NM2705 pCFJ151 Pmyo-3::mCherry::MCS::unc-10 3'UTR. The *myo-3* promoter was amplified from pCFJ104 with oligonucleotides 4220 and 4221, digested with SbfI and NotI and inserted into similarly digested NM2704 replacing the *rab-3* promoter.

NM2849 Prab-3::ptrn-1::mcherry. A full length *ptrn-1a* cDNA was amplified from first strand cDNA using oligonucleotides 4741 and 4742, digested with SbfI and NheI and inserted into NM2704.

NM2925 Pptrn1::ptrn-1::mcherry. The *ptrn-1* promoter was amplified using oligonucleotides 4828 and 4829, digested with SbfI and Not I, and inserted into NM2849 replacing the *rab-3* promoter.

NM2926 Pptrn-1::mcherry. The *ptrn-1* promoter was amplified using oligonucleotides 4828 and 4829 digested with SbfI and NotI and inserted into similarly digested NM2705.

NM2930 Pmyo-3::ptrn-1::mcherry. NM2705 and NM2849 were digested with NotI and SbfI and the *myo-3* promoter was ligated in place of the *rab-3* promoter.

NM2645 pCFJ355K *mec-7p* cherry gpd2/3 GFP-RAB-3. The *mec-7* promoter driving in mCherry and GFP-RAB-3 in an operon separated by the *gpd-2/*3 intergenic region in a neomycyin resistant derivative of the X-chromosome MosSCI integration plasmid pCFJ355 (*Frøkjær-Jensen et al., 2012*). This plasmid contains both *Cbunc-119* and neomycin selectable markers. An annotated version of the sequence of this plasmid is available at http://thalamus.wustl.edu/nonetlab/ResourcesF/seqinfo.html.

## Transgenic animals

*jsIs1077* was created by ballistic transformation (*Praitis et al., 2001*) of NM2041 pMec-7 GFP-ELKS-1 CbUNC-119 (*Kural et al., 2009*) into *unc-119(ed3)* and outcrossed into other genetic backgrounds. *jsIs1269* was created by MosSCI-mediated insertion of NM2645 into the *ttTi14024* Mos insertion site on the X chromosome using the direct integration protocol (*Frøkjær-Jensen et al., 2012*). The insertion was determined to be single copy by long range PCR and restriction digestion. *jsEx1284*, *jsEx1292*, *jsEx1295*, and *jsEx1297* were created by standard germline injections (*Mello et al., 1991*) of plasmids NM2849, NM2930, NM2925, and NM2926, respectively along with co-injection marker plasmids NM1090 (Prab-3::GFP) (*Mahoney et al., 2006*), pPD118.20 (Pmyo-3::GFP), pPD118.33(Pmyo-2::GFP) (Gifts of Andy Fire) as listed in the strain list. The plasmid of interest was injected at 30 ng/μl and co-markers at 5 ng/μl along with pBluescript KS(+) carrier DNA (150 ng/μl).

## Microscopy

Animals were anesthetized using either 10 mM sodium azide or immobilized using 0.1 micron microspheres and imaged on agarose pads (*Kim et al., 2013*). Observations of fluorescent proteins were made using an Olympus B-MAX microscope with epifluorescence. 3-D images stacks were acquired using a Ziess Axioksop equipped an ASI piezo XYZ-motorized stage, Ludl high speed electronic filter wheels and shutters, and a Hamamatsu Orca-R2 cooled CCD camera all controlled by Volocity software. Typically, images were collected using a 40X Neofluar lens, and 1 μm steps. Stacks of images were flattened using a maximal projection to visualize neuronal structures that crossed multiple individual focal planes. For immunocytochemistry, animals were fixed in 2% formaldehyde and stained with antibodies directed against GFP (mouse monoclonal, Clonetech) and MEC-7 (kindly provided by M Chalfie; *Mitani et al., 1993*) as previously described (*Schaefer et al., 2000*).

To quantify locomotion defects, L4 Animals were transferred to fresh unseeded NMG plates and imaged for 30 s using a Spot camera at 1 frame per second. The velocity of animals was quantified using worm tracker software (*Ramot et al., 2008*).

## Developmental analysis

Synchronous populations were generated from eggs laid by adult hermaphrodites over a 30-min time interval. Animals were scored by imaging *jsIs973* and *jsIs1077* using epifluorescence. Some animals were rescued from the anesthetic by being placed in a drop of M9, isolated to individual plates containing *E. coli*, and re-imaged at a later stage of development.

## Colchicine treatment

Worms were grown on NGM agar plates spotted with *E. coli* mixed with 1 mM of colchicine. L4 animals from the F2 generation were analyzed using fluorescence microscopy.

## *C. elegans* strains

The following transgenes were created for this analysis:
*jsEx1284* [Prab-3::ptrn-1:mcherry; Pmyo-2:GFP;Prab-3::GFP,Pmyo-3::GFP]
*jsEx1292* [Pmyo-3::ptrn-1:mcherry; Pmyo2::GFP]
*jsEx1297* [Pptrn-1::ptrn-1:mcherry; Pmyo-2:GFP;Prab-3::GFP,Pmyo-3::GFP],
*jsEx1295* [Pptrn-1::mcherry; Pmyo-2:GFP;Prab-3::GFP,Pmyo-3::GFP]
*jsIs1077* IV [Pmec-7::GFP::ELKS-1; Cbunc-119(+)]
*jsIs1269* X [Pmec-7::mcherry -gdp2/3 intergenic- GFP:RAB-3; Cbunc-119(+)]
Other transgenes used in this analysis include *jsIs821 X [Pmec7::rab-3:GFP; unc-119(+)]* (*Bounoutas et al., 2009*)

*jsIs973 III [Pmec7::RFP; unc-119(+)]* (**Zheng et al., 2011**)

*oxIs12 [Punc-47::GFP; lin-15(+)]* aka *osIn12* (**McIntire et al., 1997**), *nuIs25 [Pglr-1::GFP; lin-15(+)]* (**Rongo et al., 1998**)

The following strains were used in this analysis:

EG4322 *ttTi5605 II; unc-119(ed3)*

FX5597 *ptrn-1(tm5597) X*

NM4192 *jsIs973; jsIs1077; ptrn-1(js1286)*

NM4406 *jsIs973; jsIs1077; ptrn-1(tm5597)*

NM3947 *jsIs973; jsIs1077*

NM4353 *atat-2(ok2415) jsIs1269*

NM4405 *jsIs973; mec-17(ok2109)*

NM4354 *mec-17(ok2109); atat-2(ok2415) jsIs1269*

NM4544 *dlk-1(km12), jsIs973; mec-17(ok2109)*

NM4386 *jsIs973; jsIs1077; ptrn-1(js1286); jsEx1284*

NM4450 *jsIs973; ptrn-1(js1286); jsEx1292*

NM4490 *dlk-1(km12); jsIs973; rpm-1(ok364); jsIs821 ptrn-1(tm5597)*

NM4485 *jsIs973; rpm-1(ok364); jsIs821 ptrn-1(tm5597)*

NM4483 *dlk-1(km12); jsIs973; jsIs821 ptrn-1(tm5597)*

NM4482 *jsIs973; jsIs821 ptrn-1(tm5597)*

NM4484 *dlk-1(km12); jsIs973; rpm-1(ok364); jsIs821*

NM4480 *rpm-1(ok364); jsIs973; jsIs821*

NM4481 *dlk-1(km12); jsIs973; jsIs821*

NM3361 *jsIs973; jsIs821*

NM4470 *jsIs973; ptrn-1(js1286); jsEx1297*

NM4476 *jsEx1295; unc-119(ed3); ttTi5605*

NM4422 *ptrn-1(tm5597); oxIs12*

NM4419 *nuIs25; jsIs973; ptrn-1(tm5597)*

## Acknowledgements

We acknowledge the Mitani Lab and National Bioresource Project Japan for kindly providing the *ptrn-1(tm5597)* mutant nematode, Michael Crowder for the use of his worm tracker software setup and Robert Barstead for whole genome sequencing.

## Additional information

### Funding

| Funder | Grant reference number | Author |
|---|---|---|
| National Institute of Neurological Disorders and Stroke | NS040094 | Michael L Nonet |

The funder had no role in study design, data collection and interpretation, or the decision to submit the work for publication.

### Author contributions

JDM, Conception and design, Acquisition of data, Analysis and interpretation of data, Drafting or revising the article; JJC, Acquisition of data, Analysis and interpretation of data; MLN, Conception and design, Analysis and interpretation of data, Drafting or revising the article

## Additional files

### Supplementary files

• Supplementary file 1. Oligonucleotides used in this study.

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
