## [Decision Letter]

Thank you for sending your work entitled “The *C. elegans* microtubule minus-end binding homolog PTRN-1 stabilizes synapses and neurites” for consideration at *eLife*. Your article has been favorably evaluated by a Senior editor, a member of our Board of Reviewing editors, and three reviewers.

The prevailing view is that the findings described in your paper are potentially interesting, but a number of significant concerns have been raised that we ask you to address in a revised version of the manuscript:

1) The Introduction states that “we report that genetic lesions in *ptrn-1*…require *dlk-1* for regenerative morphological changes in neurons.” Similar statements about regeneration are scattered throughout the manuscript. This paper doesn't contain any data on regeneration at all. This sentence, and others like it that discuss regeneration, should be rewritten to focus on maintenance. *dlk-1* has multiple functions in neurons-in regeneration, degeneration, synapses, etc. The emphasis on regeneration seems misplaced.

2) A strength of this work is that patronin was identified in a forward screen. This could provide important context about the landscape of genes that have similar phenotypes (if any). But the screen is not well described. It would be very helpful to know how many genomes were screened, how many genes were found, and what their identity is if known.

3) mCherry is reported to be prone to aggregation. In Figure 3 mCherry puncta can be observed. The basal level of mCherry puncta makes it difficult to interpret the puncta that are observed in Figure 3. These latter figures appear to have much higher protein expression, much higher exposure, or both. Perhaps this accounts for the puncta? Also, it is stated that the fusion protein is “functional”-does this mean it rescues the mutant when expressed at reasonable levels? These data should be shown. Finally, Figure 3 is confusing for a separate reason-the mCherry appears to be localized to cell nuclei, but this is not stated in the legend or the Methods.

4) Patronin clearly functions in maintenance, but it also seems to function in development according to the data. In Figure 4, about a third of the collateral branch defects are developmental. This developmental role should be addressed in the manuscript.

5) The phenotype for patronin mutants in collateral branches is 60% wild type in Figure 4, but 13% wild type in Figure 5. This is a really big difference, despite a fairly high sample size. What might account for this?

6) The interpretation of the cholchicine experiment is incomplete. It is possible that the increased sensitivity of patronon mutants to the drug is because their microtubules are perturbed, as the manuscript suggests. But it is also possible, for example, that patronin mutants are more sensitive to sub-threshold amounts of perturbation. The acetylation experiments seek to link microtubule stability to *dlk-1*, but don't address patronin. Unless a direct experiment is done to link patronin to microtubule stability, this section needs to be revised.

7) The presentation of phenotypic data is often problematic. For example, Figure 1 shows a bar graph depicting ‘individuals with PLM overextension (%)’. There are two PLMs per animal, so it is typical and preferred to report the penetrance based on scoring cells (percentage of affected PLMs) rather than scoring individuals with PLM defects. How is an animal with two defective PLMs scored compared to only one? The ‘individual’ method inflates the penetrance if most animals have only one defective PLM and makes it impossible to compare strains if the distribution of mutant PLMs is different. In particular, the penetrance of D-type motor neuron defects (6 DD + 13 VD) is likely much lower on a cell basis compared to an ‘individual’ basis. It would also be helpful for a broader audience to label the ALM, PVM and PLM neurons in Figure 1.

8) The error bars on graphs depicting percentages of phenotypes are stated to represent standard deviation. It is unclear why a standard deviation is shown or how it was even calculated. For this type of data, the standard error of proportion would seem appropriate or a chi square analysis could be done to compare the penetrance of different strains. In particular, it is worth determining whether PTRN-1 and RPM-1 act in the same or separate pathways by comparing the penetrance of single and double mutants. However, it is unclear whether a sufficient number of animals were scored to determine whether the penetrance is different between the singles and double. In most cases, the samples sizes appear small, in part, because the n values are based on animals rather than cells.

9) It is difficult to see the details in many of the images. In particular, the quality of the images in Figure 3 is poor. It seems that maybe a NLS-GFP was used for the *ptrn-1* expression experiments; it is not hard to say whether it is nuclear or cytoplasmic. It would be nice to show that the PTRN-1-GFP puncta colocalize with MTs.

10) There are several typos and grammatical errors (e.g., Figure 1—figure supplement 2: individuals with defects is stated on both graphs, ‘Data’ is plural not singular). The authors often do not use proper *C. elegans* nomenclature, e.g., Ptrn-1 phenotypes instead of *ptrn-1* for gene and PTRN-1 for protein. There are many awkward or confusing phrases, e.g., Because the effects of colchicine treatment were additive in combination with *ptrn-1* mutants… The authors should contact the curators of wormbase to update the annotation of *ptrn-1*. At present, their *ptrn-1b* gene structure differs from wormbase version.

---

## [Author Response]

*1) The Introduction states that “we report that genetic lesions in* ptrn-1*...require* dlk-1 *for regenerative morphological changes in neurons.” Similar statements about regeneration are scattered throughout the manuscript. This paper doesn't contain any data on regeneration at all. This sentence, and others like it that discuss regeneration, should be rewritten to focus on maintenance.* dlk-1 *has multiple functions in neurons-in regeneration, degeneration, synapses, etc. The emphasis on regeneration seems misplaced*.

We agree that the manuscript speaks more to neurite remodeling than regeneration. Since we have not used a regeneration paradigm, we have refocused the text on neurite remodeling.

*2) A strength of this work is that patronin was identified in a forward screen. This could provide important context about the landscape of genes that have similar phenotypes (if any). But the screen is not well described. It would be very helpful to know how many genomes were screened, how many genes were found, and what their identity is if known*.

We have updated the manuscript to include additional information describing the screen, as requested. In short, we found four candidates in a screen of 1134 haploid genomes. The other three mutants identified in the screen remain uncharacterized.

*3) mCherry is reported to be prone to aggregation. In*
Figure 3
*mCherry puncta can be observed. The basal level of mCherry puncta makes it difficult to interpret the puncta that are observed in*
Figure 3*. These latter figures appear to have much higher protein expression, much higher exposure, or both. Perhaps this accounts for the puncta? Also, it is stated that the fusion protein is “functional”-does this mean it rescues the mutant when expressed at reasonable levels? These data should be shown. Finally,*
Figure 3
*is confusing for a separate reason-the mCherry appears to be localized to cell nuclei, but this is not stated in the legend or the Methods*.

The reviewers found the presentation of Figure 3 confusing. To clarify we have split the original Figure 3 into the new Figures 3 and 4, and additional images have been added to both figures. In the new Figure 3, images are all from a transcriptional reporter that uses the *ptrn-1* promoter to drive expression of mCherry, which is indeed primarily nuclear localized.

In the original Figure 3, images 3E-3H were from a translational reporter that uses the *ptrn-1* promoter to drive expression of a functional PTRN-1-mCherry C-terminal fusion protein. To address concerns that puncta result from aggregation of mCherry, we added new images that use the *mec-4* promoter to drive expression of a GFP-PTRN-1 N-terminal fusion protein (4D-4G). Additionally, data showing fusion protein functionality under the native *ptrn-1* promoter have been added to Figure 1, and the Figure legend notes the location of this data.

*4) Patronin clearly functions in maintenance, but it also seems to function in development according to the data. In*
Figure 4*, about a third of the collateral branch defects are developmental. This developmental role should be addressed in the manuscript*.

Figure 5 (formerly 4B) shows that a fraction of collateral branch defects occur prior to the L1 time-point. We are unable to determine if aberrant neurite morphology seen in L1 animals is due to developmental defects or to early remodeling events. The data in Figure 5 (formerly Figure 4) is now analyzed by cell rather than by animal (as requested in reviewer comment 5), and the potential developmental role for *ptrn-1* shown in Figure 5 (formerly 4B) has been discussed in the manuscript.

*5) The phenotype for patronin mutants in collateral branches is 60% wild type in*
Figure 4*, but 13% wild type in*
Figure 5*. This is a really big difference, despite a fairly high sample size. What might account for this*?

Several factors could account for the difference in collateral branching between the two figures. The figures feature different alleles js1286 (Figure 4) and tm5597 (Figure 5), were collected at different temperatures 22°C (4B) and 24°C (Figure 5), and one was collected using a synchronized population of animals at fixed time-points (Figure 4) while in the other animals were selected for analysis based on L4 morphology (5A). To address these concerns, we recollected the data for Figure 5 (currently Figure 6), using a temperature of 22°C. The current Figure 5 (formerly 4) & 6 (formerly 5) are now displayed by cell rather than by animal. Figure 5 (formerly 4B) now shows 79% wild type collateral branches and Figure 6 (formerly 5A) shows 63% wild-type branches. Additionally, the figure legends have been updated to explicitly address the differences in how animals were selected for each analysis and in the alleles used.

*6) The interpretation of the cholchicine experiment is incomplete. It is possible that the increased sensitivity of patronon mutants to the drug is because their microtubules are perturbed, as the manuscript suggests. But it is also possible, for example, that patronin mutants are more sensitive to sub-threshold amounts of perturbation. The acetylation experiments seek to link microtubule stability to* dlk-1*, but don't address patronin. Unless a direct experiment is done to link patronin to microtubule stability, this section needs to be revised*.

We updated the manuscript to acknowledge alternative interpretations of the colchicine experiment, and removed sentences that link *ptrn-1* to microtubule stability.

*7) The presentation of phenotypic data is often problematic. For example,*
Figure 1
*shows a bar graph depicting ‘individuals with PLM overextension (%)’. There are two PLMs per animal, so it is typical and preferred to report the penetrance based on scoring cells (percentage of affected PLMs) rather than scoring individuals with PLM defects. How is an animal with two defective PLMs scored compared to only one? The ‘individual’ method inflates the penetrance if most animals have only one defective PLM and makes it impossible to compare strains if the distribution of mutant PLMs is different. In particular, the penetrance of D-type motor neuron defects (6 DD + 13 VD) is likely much lower on a cell basis compared to an ‘individual’ basis. It would also be helpful for a broader audience to label the ALM, PVM and PLM neurons in*
Figure 1.

We added a panel to Figure 1—figure supplement 1 showing the variance in penetrance of overextension is less than 10% in *ptrn-1* mutants. We have used a darker font for ALM, PVM and PLM labels in Figure 1 and we labeled the neurons in Figure 1 and Figure 1—figure supplement 1 as requested. We have updated Figure 5 (formerly Figure 4) and Figure 6 (formerly Figure 5) to report phenotypic data on a “by cell” basis. While we agree with the reviewer comment that penetrance of D-type motor neuron defects would be lower on a by-cell basis compared to an individual basis, we included the quantification of these defects as a figure supplement because our purpose in analyzing motor neurons is to report that we do find defects in other neurons (aside from mechanosensory neurons that represent the core of our analysis). The text of our manuscript does not refer to the penetrance of motor neuron defects and we are happy to remove Figure 1—figure supplement 1 if this quantification remains a persistent reviewer concern.

*8) The error bars on graphs depicting percentages of phenotypes are stated to represent standard deviation. It is unclear why a standard deviation is shown or how it was even calculated. For this type of data, the standard error of proportion would seem appropriate or a chi square analysis could be done to compare the penetrance of different strains. In particular, it is worth determining whether PTRN-1 and RPM-1 act in the same or separate pathways by comparing the penetrance of single and double mutants. However, it is unclear whether a sufficient number of animals were scored to determine whether the penetrance is different between the singles and double. In most cases, the samples sizes appear small, in part, because the n values are based on animals rather than cells*.

The error bars on graphs depicting percentages of phenotypes have been changed to Standard Error of Proportion (SEP) as requested. Figures 5 and 6 have been analyzed based on a by-cell basis rather than by animal as suggested.

*9) It is difficult to see the details in many of the images. In particular, the quality of the images in*
Figure 3
*is poor. It seems that maybe a NLS-GFP was used for the* ptrn-1 *expression experiments; it is not hard to say whether it is nuclear or cytoplasmic. It would be nice to show that the PTRN-1-GFP puncta colocalize with MTs*.

Images from Figure 3 have been enlarged and higher magnification images have been used in place of the previous subpanels. Although the transcriptional fusion is not to an NLS-cherry, the mCherry does localize primarily to nuclei in this transgene. Given that the Figure 3 reporter is a transcriptional reporter, the subcellular localization is not relevant to interpretation of the figure – the signal is simply a means to identify the cell type expressing the reporter.

Additional images have also been added to Figure 4, which documents localization of a functional PTRN-1-mCherry fusion protein. In addition, we have added co-localization data showing co-localization between GFP-PTRN-1 and MEC-7 (beta tubulin) in mechanosensory neurons (Figure 4).

*10) There are several typos and grammatical errors (e.g.,*
Figure 1—figure supplement 2*: individuals with defects is stated on both graphs, ‘Data’ is plural not singular). The authors often do not use proper* C. elegans *nomenclature, e.g., Ptrn-1 phenotypes instead of* ptrn-1 *for gene and PTRN-1 for protein. There are many awkward or confusing phrases, e.g., Because the effects of colchicine treatment were additive in combination with* ptrn-1 *mutants… The authors should contact the curators of wormbase to update the annotation of* ptrn-1*. At present, their* ptrn-1b *gene structure differs from wormbase version*.

Typos: We have revised the listed typos, grammatical errors and awkward phrases, and attempted to revise similar errors and other statements that might be viewed as awkward.

Nomenclature: We are abiding to the *C. elegans* nomenclature published in Horvitz HR, Brenner S, Hodgkin J, Herman RK. (1979). A uniform genetic nomenclature for the nematode *Caenorhabditis elegans*. Molecular and General Genetics. 175 (2): 129-33. Thus, we are using italics (*ptrn-1)* when referring to the gene, Capitals (PTRN-1) when referring to the protein, and first letter only capitalized (Ptrn-1) to refer to phenotype.

Gene structure: In the case of *ptrn-1*, Wormbase is not dependable, accurate or static. Thus, we are not basing our figure on current wormbase nomenclature since it will likely change again. When we submitted in November, Wormbase showed 3 isoforms (1 of which was purely computational and erroneous, and a second that was based on an unreliable compound EST). On Jan 13, 2014, the Wormbase annotation is different and now includes both *ptrn-1* isoforms we have documented, but still includes two questionable isoforms. We have made Wormbase aware of the errors and now provide the sequence of the two isoforms as supplemental data. We would be happy to submit these to Genbank if *eLife* requests, but given the RNA-seq data in Wormbase, these forms can be deduced easily from RNA-seq and EST data.